# Excess all-cause deaths stratified by sex and age in Peru: a time series analysis during the COVID-19 pandemic

Max Carlos Ramírez-Soto ,[1,2] Gutia Ortega-Cáceres,[3]
Hugo Arroyo-Hernández [4]

¹Centro de Investigación en Salud Publica, Facultad de Medicina Humana, Universidad San Martin de Porres, Lima, Peru
²Facultad de Ciencias de la Salud, Universidad Tecnológica del Peru, Lima, Peru
³Escuela de Posgrado, Universidad Ricardo Palma, Lima, Peru
⁴Oficina General de Información y Sistemas, Instituto Nacional de Salud, Lima, Peru

**Correspondence to**
Dr Max Carlos Ramírez-Soto;
maxcrs22@gmail.com

## ABSTRACT

**Background** In this study, we estimated excess all-cause deaths and excess death rates during the COVID-19 pandemic in 25 Peruvian regions, stratified by sex and age group.

**Design** Cross-sectional study.

**Setting** Twenty-five Peruvian regions with complete mortality data.

**Participants** Annual all-cause official mortality data set from SINADEF (Sistema Informático Nacional de Defunciones) at the Ministry of Health of Peru for 2017–2020, disaggregated by age and sex.

**Main outcome measures** Excess deaths and excess death rates (observed deaths vs expected deaths) in 2020 by sex and age (0–29, 30–39, 40–49, 50–59, 60–69, 70–79 and ≥80 years) were estimated using P-score. The ORs for excess mortality were summarised with a random-effects meta-analysis.

**Results** In the period between January and December 2020, we estimated an excess of 68 608 (117%) deaths in men and 34 742 (69%) deaths in women, corresponding to an excess death rate of 424 per 100 000 men and 211 per 100 000 women compared with the expected mortality rate. The number of excess deaths increased with age and was higher in men aged 60–69 years (217%) compared with women (121%). Men between the ages of 40 and 79 years experienced twice the rate of excess deaths compared with the expected rate. In eight regions, excess deaths were higher than 100% in men, and in seven regions excess deaths were higher than 70% in women. Men in eight regions and women in one region had two times increased odds of excess death than the expected mortality. There were differences in excess mortality according to temporal distribution by epidemiological week.

**Conclusion** Approximately 100 000 excess all-cause deaths occurred in 2020 in Peru. Age-stratified excess death rates were higher in men than in women. There was strong excess in geographical and temporal mortality patterns according to region.

## INTRODUCTION

Following the outbreak of COVID-19 in China in 2019, SARS-CoV-2 spread rapidly worldwide.[1] During the weeks that followed, there was a rapid increase in the incidence and mortality rates of COVID-19 in European

## Strengths and limitations of this study

► The main strength of our study was the large number of deaths included (302 177 in men and 236 733 in women) for estimating excess all-cause deaths.

► Although our analysis was based on the proportion of mortality count excess (%) and excess death rates (per 100 000 inhabitants), our estimates of excess all-cause deaths are similar to estimates of modelled studies.

► The simplicity in the analysis of excess mortality offers an opportunity to use the findings in epidemiological surveillance and their interpretation by those responsible for formulation of public policies and health authorities at different levels.

► Although SINADEF (Sistema Informático Nacional de Defunciones) is an official database, there may have been delays in registration due to lack of notification of non-hospital or manually registered deaths.

► We present the results of a descriptive analysis and so are not able to comment on causality in excess deaths.

countries, USA and Latin American countries.[2] Previous studies on excess deaths (the gap between observed and expected deaths) during the COVID-19 pandemic found that COVID-19-related deaths and deaths from other diseases increased along with age and were higher in men 60 years of age and older compared with women.[3–10] More recently, an assessment of the direct and indirect effects of the COVID-19 pandemic on mortality revealed that approximately one million excess deaths occurred in 2020 in 29 high-income countries. Additionally, these excess death rates were higher in men than in women in almost all countries.[11]

Peru is one of the countries that have been affected the most by the COVID-19 pandemic. The first cases in Peru were reported in March 2020. In the weeks that followed, a rapidly increasing number of cases and fatalities was observed across many Peruvian regions. By the end of December 2020, about 1 million

confirmed COVID-19 cases and more than 90 000 deaths had been officially reported.[12] In Peru, data on excess all-cause deaths have been reported, but analyses have not accounted for age and sex standardisation. From 29 March 2020 (week 14), a significant rate of excess deaths in comparison with expected deaths based on the figures from 2017 to 2019 has been observed at the national level, with this rate reaching the highest level in week 32 and then decreasing.[13] This increase in all-cause deaths is concomitant with the COVID-19 pandemic, as it has previously been shown that COVID-19 mortality rate (MR) increases with age and is high in men aged 60 years and over.[14 15] Despite these findings, to date excess mortality stratified by sex, age and region is not known at a national level. Assessment of the impact of the COVID-19 pandemic on mortality in Peru should include analysis of both the direct effect of the pandemic on deaths caused by COVID-19 and the indirect effect of the pandemic on deaths unrelated to COVID-19, since patients with chronic conditions may have been turned away from the healthcare system due to concerns relating to COVID-19 infection and social and economic changes.

In this study, we estimated the excess all-cause deaths and the excess death rates during the first year of the COVID-19 pandemic in 25 Peruvian regions, stratified by sex, age group and epidemiological week.

## METHODS
### Study design
We performed a cross-sectional study following the Strengthening the Reporting of Observational Studies in Epidemiology reporting guidelines.[16] This study was a time series analysis of annual data on all-cause mortality obtained from 25 Peruvian regions and contained official, valid and complete mortality data between 2017 and 2020, disaggregated by age and sex. The Peruvian regions are first-level administrative divisions of the country.

### Data collection
We used weekly death data extracted from the National Information Technology System for Death Records (in Spanish, 'Sistema Informático Nacional de Defunciones' or SINADEF) at the Ministry of Health of Peru.[17] The source data (SINADEF) are in Spanish. We used death registers for 25 Peruvian regions from 1 January to 31 December (1–52 epidemiological weeks) and the preceding 3 years (2017–2019). SINADEF is a computer application that registers data on deaths, generates death certificates and creates a statistical report; it includes fetal deaths and deaths of unidentified persons.[17] Open access data include information stratified by age/sex, the basic cause of death by the International Classification of Diseases 10th Revision, year and geographical distribution. Other information is not available on SINADEF. Information on all-cause deaths where the underlying cause was known was available, with information on region, age group (0–29, 30–39, 40–49, 50–59, 60–69,

70–79 and 80 and over) and sex included. Records where place of residence, age, sex or year were missing were excluded.

### Patient and public involvement
We used publicly available death statistics; therefore, there was no direct patient or public involvement.

### Statistical analysis
We estimated the excess deaths and the excess death rates (per 100 000 inhabitants) according to sex and age, as well as in each region by sex. Expected deaths in 2020 were obtained from the average number of deaths over the years 2017–2019. The weekly average number of deaths by sex and age in the 3 years preceding the pandemic was the expected number of deaths. Observed deaths in 2020 were the deaths reported from 1 January to 31 December. Excess all-cause deaths during the pandemic period were estimated as the difference between observed deaths and expected deaths in 2020. To make comparisons with other countries, we measured excess proportional mortality (proportion of mortality count excess; weekly or monthly deaths in 2020) as the percentage difference between the observed and the expected number of deaths ((observed deaths–expected deaths)/expected deaths×100%),[8] by sex, age, region and epidemiological week. MR per 100 000 inhabitants for both sexes was also calculated by dividing the number of deaths per region, standardised by the estimated population of each region. We calculated the expected MR for the year 2020 and the MR observed in 2020 for both sexes according to age and region (per 100 000 inhabitants). The excess death rate was calculated using the difference between the MR observed in 2020 and the MR expected in 2020 (per 100 000 inhabitants). Population counts used to calculate the MR were obtained from projections from Peru's National Institute of Statistics and Informatics (INEI, in Spanish).[18] In addition, ORs with 95% CIs for observed mortality versus expected mortality according to sex were estimated separately for each age stratum and region. These results were summarised in a random-effects meta-analysis with individual reports weighted using the weights indicated. The Mantel-Haenszel method was used to calculate the random-effects estimates. P<0.05 was considered statistically significant. The meta-analysis was performed in RevMan V.5 (Cochrane).

## RESULTS
From 1 January to 31 December 2020, a total of 553 348 all-cause deaths occurred in Peru. Of these deaths, we included a total of 302 177 and 236 733 deaths in men and women, respectively (figure 1).

### Excess deaths and excess death rates by sex and age
Between 1 January 2020 and 31 December 2020, a total of 127 000 all-cause deaths (MR 784 deaths per 100 000 men) were reported in men. In comparison with the expected

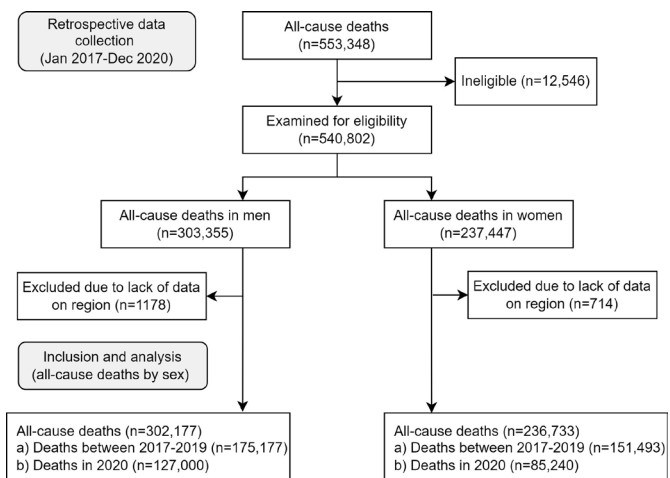

**Figure 1** Study flow chart of all-cause death excess using the STROBE reporting guidelines. STROBE, Strengthening the Reporting of Observational Studies in Epidemiology.

58 392 deaths (MR 361 per 100 000 men), the number of excess deaths during the same period was 68 608 (117%) and the excess death rate was 424 deaths per 100 000 men. During the same period, a total of 85 240 (MR 519 deaths per 100 000 women) all-cause deaths were reported in women. In comparison with the expected 50 498 deaths (MR 307 deaths per 100 000 women), the number of excess deaths during the same period was 34 742 (69%) and the excess death rate was 211 deaths per 100 000 women.

The number of excess deaths increased with age and was higher in men aged 60–69 years (217%) compared with women (121%) (figure 2A). The all-cause MR in relation to the expected death rate increased with age and was higher in individuals aged 80 years or over in both sexes (figure 2B). In addition, the increase in all-cause MR was more than twice as high in men compared with that in women.

### OR for excess death rates stratified by sex and age

Our meta-analysis shows a continuous age-dependent increase in the number of excess deaths in men and women (figure 3). Overall, men and women had 2.08

**Figure 3** OR for excess death rates stratified by age of (A) men and (B) women in Peru. OR with 95% CI estimated by age and sex for observed mortality in 2020 versus expected mortality in 2020 (average number of deaths over the years 2017–2019).

(95% CI 1.59 to 2.73) and 1.67 (95% CI 1.41 to 1.96) times increased odds of excess mortality compared with expected mortality, respectively. Men in the 40–79 years age group had two times higher odds of excess mortality. Men in the 60–69 years age group had 3.23 (95% CI 3.15 to 3.31) times increased odds of excess mortality, while

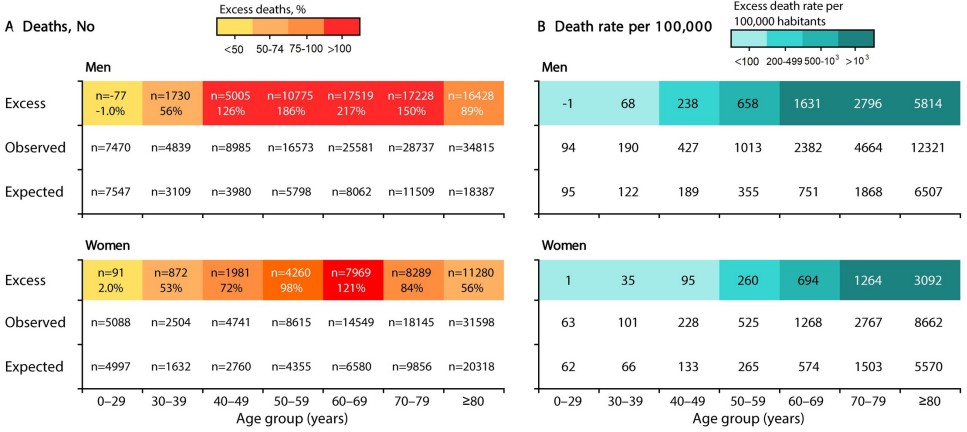

**Figure 2** Excess number of (A) deaths and (B) excess death rates in men and women in Peru stratified by age.

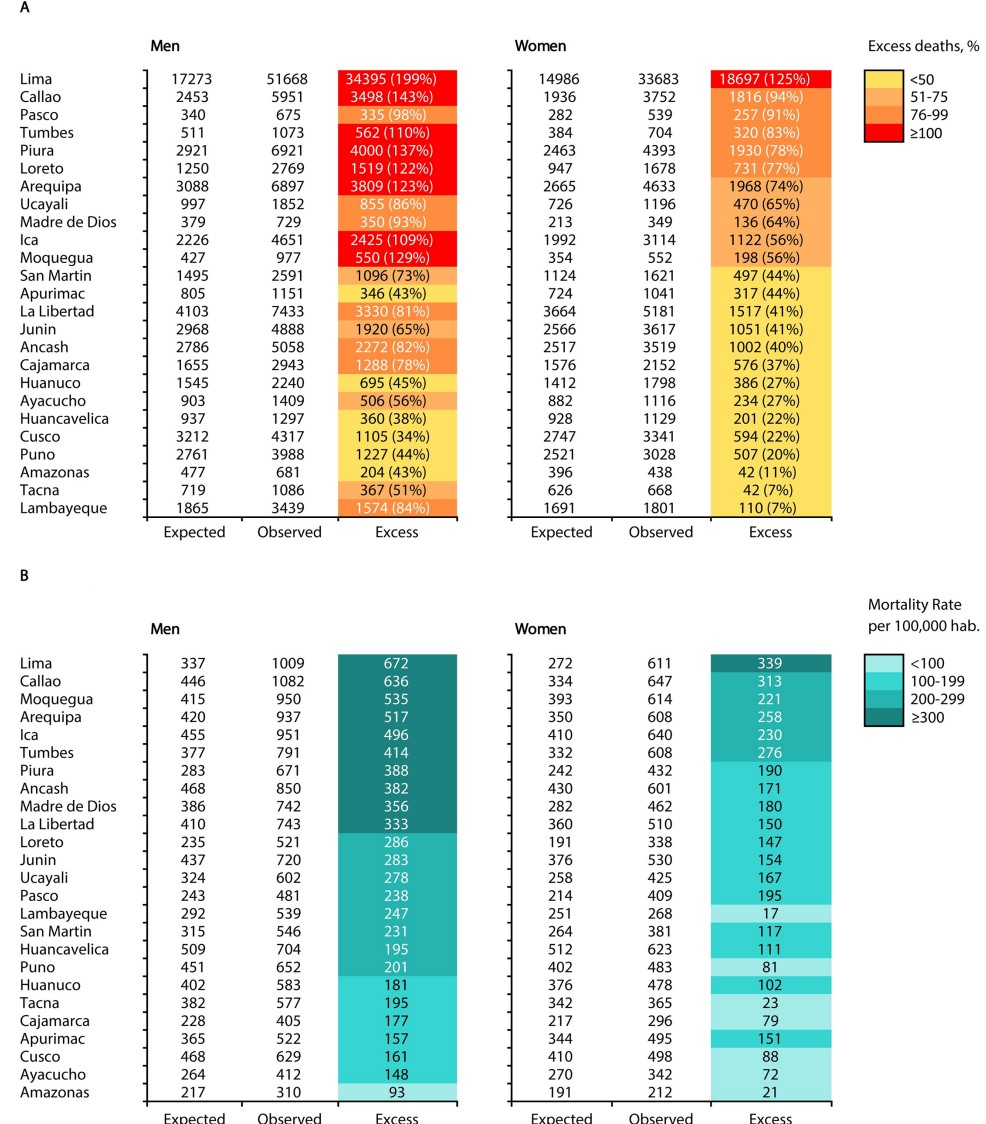

**Figure 4** (A) Observed, expected and excess deaths and (B) excess death rates in men and women stratified by region in Peru.

women had 2.23 (95% CI 2.16 to 2.29) times increased odds (figure 3A,B).

### Excess deaths and excess death rates by sex and region

The excess all-cause death rate relative to the expected death rate varied between regions. For men, the excess death rate varied from 34% in Cusco to 199% in Lima, while for women it varied from 7% in Tacna and 125% in Lima. The five regions with the highest excess death rates in men were Lima (199%), Callao (143%), Piura (137%), Moquegua (129%) and Arequipa (123%); the five regions with the highest excess deaths in women were Lima (125%), Callao (94%), Paso (91%), Tumbes (83%) and Piura (78%) (figure 4A). The highest excess death rates in men (per 100 000 men) and women (per 100 000 women) occurred in Lima, Callao, Moquegua, Arequipa, Ica and Tumbes (figure 4B).

### OR for excess death rates stratified by sex and region

Our meta-analysis shows a continuous increase in the rate of excess deaths in men and women (figure 5). Men in Lima had 3.01 (95% CI 2.96 to 3.06) times increased odds of excess mortality. Men in Callao, Piura, Arequipa, Loreto, Tumbes and Moquegua had two times increased odds of excess mortality compared with expected mortality (figure 5A), while only in Lima did women have 2.26 (95% CI 2.21 to 2.30) times increased odds of excess mortality compared with expected mortality (figure 5B).

### Temporal distribution of excess deaths by sex and region

Figure 6 shows that the rate of excess deaths increased at the beginning of March and reached a peak at the end of the months of May and August. There were different peaks among different regions (figure 7A,B). Some regions showed staggering increases, with rates of excess deaths reaching 500% in men in Loreto in week 18

**A**

| Region | Odds Ratio | OR 95% CI |
|---|---|---|
| Amazonas | | 1.43 [1.27 , 1.61] |
| Ancash | | 1.82 [1.74 , 1.91] |
| Apurimac | | 1.43 [1.31 , 1.57] |
| Arequipa | | 2.25 [2.15 , 2.34] |
| Ayacucho | | 1.56 [1.44 , 1.70] |
| Cajamarca | | 1.78 [1.68 , 1.89] |
| Callao Province | | 2.44 [2.33 , 2.56] |
| Cusco | | 1.35 [1.29 , 1.41] |
| Huancavelica | | 1.39 [1.27 , 1.51] |
| Huanuco | | 1.45 [1.36 , 1.55] |
| Ica | | 2.10 [2.00 , 2.21] |
| Junin | | 1.65 [1.58 , 1.73] |
| La Libertad | | 1.82 [1.75 , 1.89] |
| Lambayeque | | 1.85 [1.75 , 1.96] |
| Lima | | 3.01 [2.96 , 3.06] |
| Loreto | | 2.22 [2.08 , 2.38] |
| Madre de Dios | | 1.93 [1.70 , 2.19] |
| Moquegua | | 2.30 [2.05 , 2.58] |
| Pasco | | 1.99 [1.75 , 2.27] |
| Piura | | 2.38 [2.28 , 2.48] |
| Puno | | 1.45 [1.38 , 1.52] |
| San Martin | | 1.74 [1.63 , 1.85] |
| Tacna | | 1.51 [1.38 , 1.66] |
| Tumbes | | 2.11 [1.90 , 2.34] |
| Ucayali | | 1.86 [1.72 , 2.01] |
| | | |
| Total (95% CI) | | 1.83 [1.63 , 2.06] |

Heterogeneity: Tau² = 0.09; Chi² = 3033.65, df = 24 (P < 0.00001); I² = 99%
Test for overall effect: Z = 10.06 (P < 0.00001)

**B**

| Region | Odds Ratio | OR 95% CI |
|---|---|---|
| Amazonas | | 1.11 [0.97 , 1.27] |
| Ancash | | 1.40 [1.33 , 1.47] |
| Apurimac | | 1.44 [1.31 , 1.58] |
| Arequipa | | 1.74 [1.66 , 1.83] |
| Ayacucho | | 1.27 [1.16 , 1.38] |
| Cajamarca | | 1.37 [1.28 , 1.46] |
| Callao Province | | 1.94 [1.84 , 2.05] |
| Cusco | | 1.22 [1.16 , 1.28] |
| Huancavelica | | 1.22 [1.12 , 1.33] |
| Huanuco | | 1.27 [1.19 , 1.37] |
| Ica | | 1.57 [1.48 , 1.66] |
| Junin | | 1.41 [1.34 , 1.49] |
| La Libertad | | 1.42 [1.36 , 1.48] |
| Lambayeque | | 1.07 [1.00 , 1.14] |
| Lima | | 2.26 [2.21 , 2.30] |
| Loreto | | 1.77 [1.64 , 1.92] |
| Madre de Dios | | 1.64 [1.38 , 1.95] |
| Moquegua | | 1.56 [1.37 , 1.79] |
| Pasco | | 1.92 [1.66 , 2.21] |
| Piura | | 1.79 [1.70 , 1.88] |
| Puno | | 1.20 [1.14 , 1.27] |
| San Martin | | 1.44 [1.34 , 1.56] |
| Tacna | | 1.07 [0.96 , 1.19] |
| Tumbes | | 1.84 [1.62 , 2.08] |
| Ucayali | | 1.65 [1.50 , 1.81] |
| | | |
| Total (95% CI) | | 1.47 [1.33 , 1.64] |

Heterogeneity: Tau² = 0.07; Chi² = 1911.23, df = 24 (P < 0.00001); I² = 99%
Test for overall effect: Z = 7.20 (P < 0.00001)

**Figure 5** OR for excess death rates in (A) men and (B) women according to region in Peru. OR with 95% CI estimated by region and sex for observed mortality in 2020 versus expected mortality in 2020 (average number of deaths over the years 2017–2019).

(figure 7A). The rate of excess deaths decreased by week 52 (figure 7A,B).

## DISCUSSION
### Main findings
During the COVID-19 pandemic, Peru experienced one of the highest excess all-cause MRs in the world, along with Bulgaria, North Macedonia and Serbia.[19] Excess all-cause mortality is recognised as a robust and comparable indicator of mortality associated with the COVID-19 pandemic.[20] Some countries have reported that approximately two-thirds of their excess deaths were reported as COVID-19-related deaths.[21] However, in countries such as Peru, where the effects of the pandemic have exceeded the reporting systems for mortality related to COVID-19, the measurement of excess mortality can help guide public health decisions and actions.

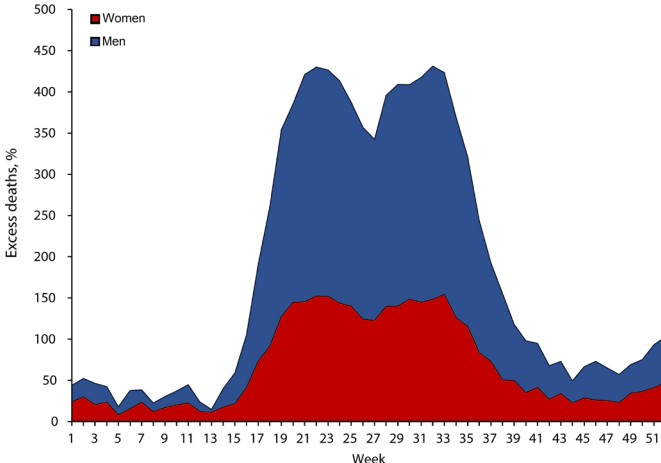

**Figure 6** Temporal distribution of excess deaths (%) in men and women in Peru.

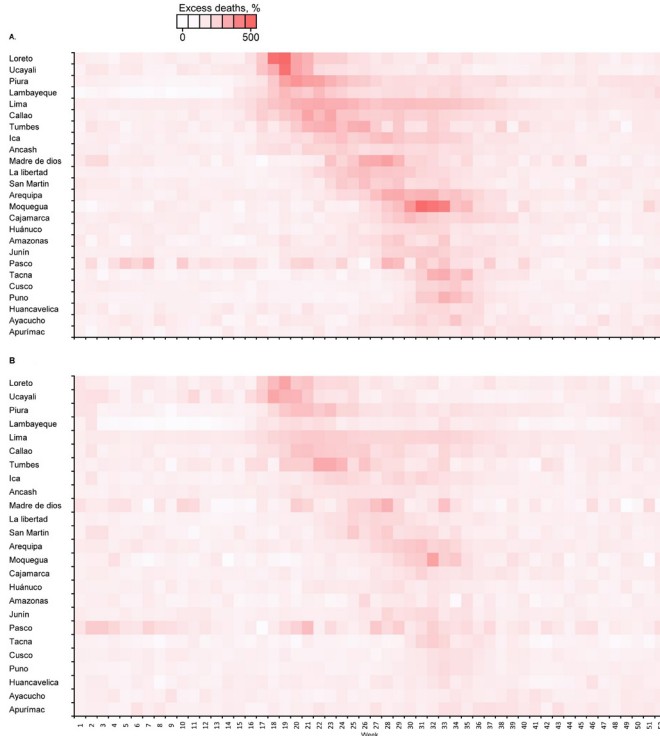

**Figure 7** Temporal distribution of excess deaths in (A) men and (B) women according to region in Peru.

## Potential explanations and implications

According to our analysis, men in Peru had an excess MR almost double that of women. This differs from the data reported in most industrialised and high-income countries, where no great difference is evident. For example, the excess mortality in Peruvian men (117%) was double that reported for England and Wales (63%), which together with Spain were the countries that experienced the highest excess mortality in Europe[20 22]; this is also higher than the rate reported in other countries of the American continent, such as Mexico (51%) and Brazil (61%).[8 23] Excess mortality in men is likely related to the COVID-19 pandemic, as SARS-CoV-2 has greater transmissibility in men, which is mainly due to their social and cultural behaviours. It has also been proposed that women have a better protective factor immune response due to their higher number of B lymphocytes and the higher intensity of their inflammatory response against viral diseases.[24 25] Because COVID-19 is a new infectious disease, the development and viability of memory T cells in men and women are still unknown, especially in the face of viral mutation. Therefore, further studies could define the sex-differential role of T cells in acute disease. Sex differences in immunopathogenesis could inform the mechanisms of COVID-19 and identify points for treatment and increase vaccine efficacy to target parts of the virus that are less likely to mutate, supported by genome analysis.[26 27]

Several previous studies have reported age-specific excess deaths. In these studies, there is wide variability in the estimation of excess mortality by sex and age.[8 28 29] As in Peru, other countries have also reported that excess in observed mortality among younger age groups was similar to or lower than the expected levels.[28] This is due to the lower number of unintentional injuries occurring due to COVID-19 containment measures, such as occupational or automobile accidents, the latter being the main cause of years of life lost due to premature death.[30 31] In our study, the excess deaths (%) varied markedly in the 50–59, 60–69 and 70–79 years age groups. The rate of excess deaths also varied with age and was higher in men aged 60–69, 70–79 and ≥80 years, in contrast to other European countries such as Belgium, the Czech Republic, Hungary, Poland and Scotland, where the excess death rates were higher in men 55–64 years of age.[28] As in Peru, in the overall US population the excess deaths in individuals 65 years and over were higher.[29] In our study, the ORs for excess mortality occurred in a wider range of age groups for men (40–79 years old); this may be related to the high prevalence of chronic diseases in adult men, which increases the likelihood of death due to COVID-19.[32 33] Thus, a recent study showed that as obesity prevalence increases, the COVID-19 MR increases in the Peruvian population ≥15 years.[34] These factors may have contributed to a higher estimated excess all-cause death rate in men than women in the ≥60 years age group in Peru.

Overall, there were different peaks of excess mortality among different regions. However, men were found to be twice as likely to die in more regions than women. Regions along the coast (Lima, Callao, Piura and Tumbes), in the Andes (Arequipa and Moquegua) and in the Amazon (Loreto) have large cities with high levels of commercial, migratory and tourist exchange. In these regions with large cities, the rates of excess death in men increased at the beginning of March, reached a peak in weeks 18 and 32, and then decreased. This increase in all-cause deaths in these regions was temporal and concomitant with the first wave of the COVID-19 pandemic in Peru. The temporal distribution was similar in women figure 7A,B. In contrast, the regions of the Andes located at higher altitudes had lower MR. This has been related to the physiological adaptation of this population to a hypoxic environment, which may have protected them from the severe impact of acute infection caused by SARS-CoV-2; in addition, these areas have a lower prevalence of diabetes, obesity and hypertension.[35] Therefore, the excess all-cause deaths are likely to be a direct effect of the COVID-19 pandemic. These characteristics, together with low compliance with social distancing measures, the closure of economic activities during the pandemic, the introduction of new variants of SARS-CoV-2 with greater transmissibility, the difficulty of early diagnosis, the collapse of hospitals and the shortage of medical oxygen, could explain the higher excess mortality in Peru.[36 37] Changes in population (male and female) behaviour brought about by lockdown measures could also have had effects on excess deaths and MR. Finally, during the country-wide lockdown implemented in Peru, access to medical and surgical care was limited and interrupted across primary and secondary prevention programmes. Further studies could evaluate the indirect effects of confinement and excess deaths and mortality by disaggregating data by chronic or acute diseases.

## Weaknesses of this study

Our study has several important limitations. First, although SINADEF is an official database, there may have been delays in registration due to lack of notification of non-hospital or manually registered deaths. Therefore, the number of registered excess all-cause deaths might have been underestimated.

A second limitation is related to the selection of expected deaths; a study of excess mortality during the pandemic carried out in 103 countries mentions that 20 of them, in which Peru is located, have a death record of less than 90% according to the Demographic Yearbook of the United Nations for 2019,[19] although there was an increasing trend in the coverage of registered deaths according to data from 2012 (70%) to 2016 (80%). In 2017, improvements to the SINADEF of Peru—with the collaboration of international institutions that allowed the increase in the use of death certificates online and the improvement in the quality of information through training, standardisation and monitoring of civil registration and vital statistics processes—shortened the gap between coverage of registered and estimated deaths.[38] However, it is likely that an increase in the registration of

deaths expected during 2017–2019 is related to coverage of deaths; therefore, the use of an average calculated in epidemiological weeks would be justified.

Third, comparison with other studies may be limited, where excess mortality uses complex models that apply periodic splines or Fourier harmonics to smooth and homogenise possible fluctuations in expected deaths and standardise their measurements.[39] We used the proportion of mortality count excess and a standard measure (P-score), which is simple and easy to replicate in any context and similar to that carried out in other studies.[8 29] Moreover, while other studies estimated the expected deaths as the average number of deaths between 2015 and 2019, we only used the 3 years preceding the pandemic (2017–2019). This method could also be easily replicated in each region without losing the opportunity to analyse important variables[8]; however, it was only possible to access 3 years as a baseline. Future analysis of excess deaths in 2021 could not include 2020 in the average expected deaths due to high death rates.

## Strengths of this study

The main strength of our study is that the developed method allows for transparency and reproducibility of the findings. The baseline for expected deaths made it possible to capture the weekly variation of observed deaths and is self-consistent with that reported in more complex methodologies.[39] The simplicity in the analysis of excess mortality allows for the opportunity to use the findings in epidemiological surveillance and their interpretation by those responsible for formulation of public policies and health authorities at different levels.

## Implications for policymakers

The findings add to the body of evidence about our understanding of excess mortality in Peru. Although the initiation of vaccinations against COVID-19 and improvements in the capacity to diagnose and care for critical patients will reduce mortality, policymakers must evaluate future measures and interventions based on gender. On the other hand, coordination initiatives between those responsible for epidemiological surveillance, vital event registration systems and diagnosis have contributed to obtaining more realistic numbers, but evaluating the interoperability of computer and statistical systems is necessary for future measurements of observed deaths and excess mortality. Finally, having up-to-date information provides government and public health officials with adequate mechanisms to track the impact of the pandemic.

## CONCLUSION

Approximately 100 000 excess deaths occurred in men and women during the COVID-19 pandemic in Peru. The age-stratified excess death rates were higher in men than in women, and men between the ages of 40 and 79 years experienced twice the rate of excess deaths compared

with the rate expected. Strong excess in geographical and temporal mortality patterns by region was found in both men and women. These findings reveal the impact of the COVID-19 pandemic on all-cause mortality up to the point where vaccination against SARS-CoV-2 started to become available in Peru.

**Contributors** MCR-S (lead author and guarantor) designed the study and oversaw the acquisition of data. MCR-S, GO-C and HA-H performed the data collection and interpretation of data. MCR-S and HA-H verified the coded results. MCR-S and HA-H analysed the data. MCR-S, GO-C and HA-H contributed to drafting the manuscript. MCR-S, GO-C and HA-H were involved in critical revisions of the manuscript for important intellectual content. All authors (MCR-S, GO-C, and HA-H) reviewed, commented and edited the manuscript and approved the final version.

**Funding** The authors have not declared a specific grant for this research from any funding agency in the public, commercial or not-for-profit sectors.

**Competing interests** None declared.

**Patient and public involvement** Patients and/or the public were not involved in the design, or conduct, or reporting, or dissemination plans of this research.

**Patient consent for publication** Not required.

**Ethics approval** This study does not involve human participants and weekly death data were obtained from SINADEF. On SINADEF, all open access data are fully anonymous and are published as part of routine surveillance (https://www.datosabiertos.gob.pe/dataset/informaci%C3%B3n-de-fallecidos-del-sistema-inform%C3%A1tico-nacional-de-defunciones-sinadef-ministerio). Therefore, this study was exempt from review by an ethics board.

**Provenance and peer review** Not commissioned; externally peer reviewed.

**Data availability statement** Data are available in a public, open access repository. Data used are freely available and can be accessed from Peruvian Ministry of Health (SINADEF, National Information System of Deaths: https://www.datosabiertos.gob.pe/dataset/informaci%C3%B3n-de-fallecidos-del-sistema-inform%C3%A1tico-nacional-de-defunciones-sinadef-ministerio) and the National Institute of Statistics and Informatics.

**ORCID iDs**
Max Carlos Ramírez-Soto http://orcid.org/0000-0003-0471-6746
Hugo Arroyo-Hernández http://orcid.org/0000-0001-5128-7820

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
