## [Reviewer comments · BMJ Open]

ARTICLE DETAILS

TITLE (PROVISIONAL)	Excess all-cause deaths stratified by sex and age in Peru: a time series analysis during the COVID-19 pandemic
AUTHORS	Ramírez-Soto, Max Carlos; Ortega-Cáceres, Gutia; Arroyo-Hernández, Hugo

VERSION 1 – REVIEW

REVIEWER	Heuveline, Patrick UCLA, CCPR
REVIEW RETURNED	03-Oct-2021

GENERAL COMMENTS	This paper analyses excess mortality from Covid-19 in one of the most affected countries to date, Peru. This fills an important gap because most studies of excess mortality during the pandemic have focused on Europe and North America, a data-driven choice that does not, or no longer at least, reflect the severity of the pandemic worldwide. The paper is clearly written, but the methods section might be a little too terse to fully understand how excess mortality was estimated. Page 5, line 25, “deaths expected based on data from the years 2017-2019” could mean different things. Did the authors average the number of deaths over those three years, or did they use another approach to “predict” counterfactual 2020 deaths from 2017-2019 data? Relatedly, it is important to note that since there are different approaches to estimate excess mortality, confidence intervals derived from one approach under-estimate the actual uncertainty (see Medrxiv preprint by Nepomuceno et al. on the sensitivity of excess mortality estimates) The following are additional suggestions that might broaden the audience for this paper: Taking age distribution into consideration, as the authors do, is clearly essential, but it is not always possible. Therefore, there is considerable interest in potential similarities and differences in the age pattern of Covid-19 mortality across populations. Can the authors shed some light on this by comparing how excess mortality rates increase with age in Peru and in the European and North American populations where excess mortality has been amply documented? To help make sense of potential differences, can the authors assess the quality of age reporting in Peru? For readers not that familiar with Peru, regional differences might be difficult to interpret. The first half of the paragraph starting on p.7, line 25, is useful in that regard, but may still not be enough to
--

	make sense of results for 25 Provinces reported in some of the Figures (3, 4, and 6). Since the discussion focuses on differences between Provinces with large commercial centers and high-altitude Provinces in the Andes, grouping Provinces in that manner may make these points more readily visible in the visual representations of the results. In any case, higher mortality in large cities with higher levels of commercial and tourist exchange has been widely observed elsewhere, but this might be only temporary (e.g., New York metropolitan area in the US). It would be interesting to report whether in Peru, these regional differences have persisted over the year.
--	--

REVIEWER	Quintana, Hedley Instituto Conmemorativo Gorgas de Estudios de la Salud, Research and Health Technology Assessment
REVIEW RETURNED	01-Nov-2021

GENERAL COMMENTS	In my view, it is an important topic to assess. However, I have some methodology concerns. Firstly, SINADEF is the website where the data comes from is in Spanish language. I am Spanish native speaker, but I am sure some readers don't speak it; therefore the authors ought warn that the source data in such language. Secondly, this study ought to fit in any of the classical observational epidemiology study designs: it looks like a collection of several cross-sectional studies where you look for secular trends. I am pretty sure the authors do know about the mortality before the study takes place: it gives a hint that we are looking a cross-sectional study. This information is needed in the title, as well as, for the correct filing of the items 6, 12 and 15 of the STROBE checklist Thirdly, according to the point STROBE, a participation flowchart is expected to be shown by the authors on page 5. For this kind of studies, it means a clean-up of the data where the manuscript relies on. It means that the authors excluded participants: it mean that the data was with x amount of records for males, y for females and z for cases with undetermined sex. After the clean-up 127,000 deaths were reported in men and 68,608 cases in females. Can the authors report the values for x, y, x-127000, y-68608 and z in a flowchart as STROBE requires? Fourthly, it is not clear how the authors calculated the expected deaths as well as the expected population for 2020. Furthermore, why does the expected population is different from the observed population, ergo the denominator? (shown in figure 2), Fifthly, the readers do not know about the "regions" of Peru. Are the "regions" a first-level administrative divisions of the country? or are they health regions? In my country, we have provinces and indigenous shires which are first-level administrative divisions and health regions: they correspond each other, except for a large province that have 4 health regions. However, the authors use the word "department" in the abstract and in figure 4. Please be consistent when are using words for describing geographical region thru the manuscript, including figures.
--

	Sixthly, I don't understand where the weights come from or whether they are needed. The only need I see for using weights in their results is for using an indirect standardization, for example 2005-2025 World population to get rid of the effect of different age groups among different populations. Finally, the authors refer to the ICD-10 as a means for coding the cause of the death. Why is this information relevant?, given that they report all-causes of death in their results and not a particular one. Reviewer #1. This paper analyses excess mortality from Covid-19 in one of the most affected countries to date, Peru. This fills an important gap because most studies of excess mortality during the pandemic have focused on Europe and North America, a data-driven choice that does not, or no longer at least, reflect the severity of the pandemic worldwide. Response: Thank you for your comment. Comment 1: The paper is clearly written, but the methods section might be a little too terse to fully understand how excess mortality was estimated. Page 5, line 25, "deaths expected based on data from the years 2017-2019" could mean different things. Did the authors average the number of deaths over those three years, or did they use another approach to "predict" counterfactual 2020 deaths from 2017-2019 data? Relatedly, it is important to note that since there are different approaches to estimate excess mortality, confidence intervals derived from one approach under-estimate the actual uncertainty (see Medrxiv preprint by Nepomuceno et al. on the sensitivity of excess mortality estimates). Response 1: Thank you for your comments. We measure excess mortality using P-score [(Observed deaths–Expected deaths)/Expected deaths*100%], i.e., the percentage difference between the observed (weekly or monthly deaths in 2020) and expected number of deaths in 2020 (the average number of deaths in the same period over the years 2017–2019) by sex, age, region and epidemiological week. Mortality rates per 100,000 inhabitants for both sexes were also calculated by dividing the number of deaths per region, standardized by the estimated population of each region. We calculated the expected mortality rate for the year 2020 (the average number of deaths in the same period over the years 2017–2019) and the mortality rate observed in 2020 for both sexes according to age and region (per 100,000 inhabitants). The excess death rate was calculated using the difference between the mortality rate observed in 2020 and the mortality rate expected in 2020 (based on the years 2017–2019). P-score is a standard measure, simple and easy to replicate in any context. It does not use 95% CI. Comment 2: Taking age distribution into consideration, as the authors do, is clearly essential, but it is not always possible. Therefore, there is considerable interest in potential similarities and differences in the age pattern of Covid-19 mortality across populations. Can the authors shed some light on this by comparing how excess mortality rates increase with age in Peru and in the European and North American populations where excess mortality has been amply documented? Response 3: Thank you for your comments. Several previous studies have reported age specific excess deaths. In these
--	---

studies, there is a wide variability in the estimation of excess mortality by sex and age (see Discussion section).
Comment 3: To help make sense of potential differences, can the authors assess the quality of age reporting in Peru?
Response 3: Thank you for your comments. In Peru, there was an increase in the use of death certificates online, and the improvement in the quality of the information (including the age reporting), through training, standardization and monitoring of civil registration and vital statistics processes, shortening the gap between the coverage of registered and estimated deaths (see Discussion section).

Comment 4: For readers not that familiar with Peru, regional differences might be difficult to interpret. The first half of the paragraph starting on p.7, line 25, is useful in that regard, but may still not be enough to make sense of results for 25 Provinces reported in some of the Figures (3, 4, and 6). Since the discussion focuses on differences between Provinces with large commercial centers and high-altitude Provinces in the Andes, grouping Provinces in that manner may make these points more readily visible in the visual representations of the results. In any case, higher mortality in large cities with higher levels of commercial and tourist exchange has been widely observed elsewhere, but this might be only temporary (e.g., New York metropolitan area in the US). It would be interesting to report whether in Peru, these regional differences have persisted over the year.
Response 4: Thank you for your comments. It was corrected (see Discussion section)

Reviewer #2. Dr. Hedley Quintana, Instituto Conmemorativo Gorgas de Estudios de la Salud

In my view, it is an important topic to assess. However, I have some methodology concerns.

Comment 1: Firstly, SINADEF is the website where the data comes from is in Spanish language. I am Spanish native speaker, but I am sure some readers don't speak it; therefore the authors ought warn that the source data in such language.

Response 1: Thank you for your comments. It was corrected (see Methods section)

Comment 2: Secondly, this study ought to fit in any of the classical observational epidemiology study designs: it looks like a collection of several cross-sectional studies where you look for secular trends. I am pretty sure the authors do know about the mortality before the study takes place: it gives a hint that we are looking a cross-sectional study. This information is needed in the title, as well as, for the correct filing of the items 6, 12 and 15 of the STROBE checklist.

Response 2: Thank you for your comments. We included the design study (see Methods section).

Comment 3: Thirdly, according to the point STROBE, a participation flowchart is expected to be shown by the authors on page 5. For this kind of studies, it means a clean-up of the data where the manuscript relies on. It means that the authors excluded participants: it mean that the data was with x amount of records for males, y for females and z for cases with undetermined sex. After

	the clean-up 127,000 deaths were reported in men and 68,608 cases in females. Can the authors report the values for x, y, x-127000, y-68608 and z in a flowchart as STROBE requires? Response 3: Thank you for your comments. We included the Figure 1. The study flow chart of all-cause death excess with the STROBE. Comment 4: Fourthly, it is not clear how the authors calculated the expected deaths as well as the expected population for 2020. Furthermore, why does the expected population is different from the observed population, ergo the denominator? (shown in figure 2). Response 4: Thank you for your comments. It was corrected (see Figure 3 and 5) Comment 5: Fifthly, the readers do not know about the "regions" of Peru. Are the "regions" a first-level administrative divisions of the country? or are they health regions? In my country, we have provinces and indigenous shires which are first-level administrative divisions and health regions: they correspond each other, except for a large province that have 4 health regions. However, the authors use the word "department" in the abstract and in figure 4. Please be consistent when are using words for describing geographical region thru the manuscript, including figures. Response 5: Thank you for your comments. It was corrected. The Peruvian regions are first-level administrative divisions of the country (see Methods section). Comment 6: Sixthly, I don't understand where the weights come from or whether they are needed. The only need I see for using weights in their results is for using a indirect standardization, for example 2005-2025 World population to get rid of the effect of different age groups among different populations. Response 6: Thank you for your comments. It was corrected (see Figures 3A,B and 5A,B) Comment 7: Finally, the authors refer to the ICD-10 as a means for coding the cause of the death. Why is this information relevant?, given that they report all-causes of death in their results and not a particular one. Response 7: Thank you for your comments. It was corrected (see Methods section).
--	---

VERSION 1 – AUTHOR RESPONSE

Reviewer #1. This paper analyses excess mortality from Covid-19 in one of the most affected countries to date, Peru. This fills an important gap because most studies of excess mortality during the pandemic have focused on Europe and North America, a data-driven choice that does not, or no longer at least, reflect the severity of the pandemic worldwide.

Response: Thank you for your comment.

Comment 1: The paper is clearly written, but the methods section might be a little too terse to fully understand how excess mortality was estimated. Page 5, line 25, “deaths expected based on data from the years 2017-2019” could mean different things. Did the authors average the number of deaths over those three years, or did they use another approach to “predict” counterfactual 2020 deaths from 2017-2019 data? Relatedly, it is important to note that since there are different approaches to

estimate excess mortality, confidence intervals derived from one approach under-estimate the actual uncertainty (see Medrxiv preprint by Nepomuceno et al. on the sensitivity of excess mortality estimates).

Response 1: Thank you for your comments. We measure excess mortality using P-score $[(\text{Observed deaths} - \text{Expected deaths}) / \text{Expected deaths} * 100\%]$, i.e., the percentage difference between the observed (weekly or monthly deaths in 2020) and expected number of deaths in 2020 (the average number of deaths in the same period over the years 2017–2019) by sex, age, region and epidemiological week. Mortality rates per 100,000 inhabitants for both sexes were also calculated by dividing the number of deaths per region, standardized by the estimated population of each region. We calculated the expected mortality rate for the year 2020 (the average number of deaths in the same period over the years 2017–2019) and the mortality rate observed in 2020 for both sexes according to age and region (per 100,000 inhabitants). The excess death rate was calculated using the difference between the mortality rate observed in 2020 and the mortality rate expected in 2020 (based on the years 2017–2019). P-score is a standard measure, simple and easy to replicate in any context. It does not use 95% CI.

Comment 2: Taking age distribution into consideration, as the authors do, is clearly essential, but it is not always possible. Therefore, there is considerable interest in potential similarities and differences in the age pattern of Covid-19 mortality across populations. Can the authors shed some light on this by comparing how excess mortality rates increase with age in Peru and in the European and North American populations where excess mortality has been amply documented?

Response 3: Thank you for your comments. Several previous studies have reported age specific excess deaths. In these studies, there is a wide variability in the estimation of excess mortality by sex and age (see Discussion section).

Comment 3: To help make sense of potential differences, can the authors assess the quality of age reporting in Peru?

Response 3: Thank you for your comments. In Peru, there was an increase in the use of death certificates online, and the improvement in the quality of the information (including the age reporting), through training, standardization and monitoring of civil registration and vital statistics processes, shortening the gap between the coverage of registered and estimated deaths (see Discussion section).

Comment 4: For readers not that familiar with Peru, regional differences might be difficult to interpret. The first half of the paragraph starting on p.7, line 25, is useful in that regard, but may still not be enough to make sense of results for 25 Provinces reported in some of the Figures (3, 4, and 6). Since the discussion focuses on differences between Provinces with large commercial centers and high-altitude Provinces in the Andes, grouping Provinces in that manner may make these points more readily visible in the visual representations of the results. In any case, higher mortality in large cities with higher levels of commercial and tourist exchange has been widely observed elsewhere, but this might be only temporary (e.g., New York metropolitan area in the US). It would be interesting to report whether in Peru, these regional differences have persisted over the year.

Response 4: Thank you for your comments. It was corrected (see Discussion section)

Reviewer #2. Dr. Hedley Quintana, Instituto Conmemorativo Gorgas de Estudios de la Salud

In my view, it is an important topic to assess. However, I have some methodology concerns.

Comment 1: Firstly, SINADEF is the website where the data comes from is in Spanish language. I am Spanish native speaker, but I am sure some readers don't speak it; therefore the authors ought warn that the source data in such language.

Response 1: Thank you for your comments. It was corrected (see Methods section)

Comment 2: Secondly, this study ought to fit in any of the classical observational epidemiology study designs: it looks like a collection of several cross-sectional studies where you look for secular trends. I am pretty sure the authors do know about the mortality before the study takes place: it gives a hint that we are looking a cross-sectional study. This information is needed in the title, as well as, for the correct filing of the items 6, 12 and 15 of the STROBE checklist.

Response 2: Thank you for your comments. We included the design study (see Methods section).

Comment 3: Thirdly, according to the point STROBE, a participation flowchart is expected to be shown by the authors on page 5. For this kind of studies, it means a clean-up of the data where the manuscript relies on. It means that the authors excluded participants: it mean that the data was with x amount of records for males, y for females and z for cases with undetermined sex. After the clean-up 127,000 deaths were reported in men and 68,608 cases in females. Can the authors report the values for x, y, x-127000, y-68608 and z in a flowchart as STROBE requires?

Response 3: Thank you for your comments. We included the Figure 1. The study flow chart of all-cause death excess with the STROBE.

Comment 4: Fourthly, it is not clear how the authors calculated the expected deaths as well as the expected population for 2020. Furthermore, why does the expected population is different from the observed population, ergo the denominator? (shown in figure 2).

Response 4: Thank you for your comments. It was corrected (see Figure 3 and 5)

Comment 5: Fifthly, the readers do not know about the "regions" of Peru. Are the "regions" a first-level administrative divisions of the country? or are they health regions? In my country, we have provinces and indigenous shires which are first-level administrative divisions and health regions: they correspond each other, except for a large province that have 4 health regions. However, the authors use the word "department" in the abstract and in figure 4. Please be consistent when are using words for describing geographical region thru the manuscript, including figures.

Response 5: Thank you for your comments. It was corrected. The Peruvian regions are first-level administrative divisions of the country (see Methods section).

Comment 6: Sixthly, I don't understand where the weights come from or whether they are needed. The only need I see for using weights in their results is for using a indirect standardization, for example 2005-2025 World population to get rid of the effect of different age groups among different populations.

Response 6: Thank you for your comments. It was corrected (see Figures 3A,B and 5A,B)

Comment 7: Finally, the authors refer to the ICD-10 as a means for coding the cause of the death. Why is this information relevant?, given that they report all-causes of death in their results and not a particular one.

Response 7: Thank you for your comments. It was corrected (see Methods section).

VERSION 2 – REVIEW

REVIEWER	Quintana, Hedley Instituto Conmemorativo Gorgas de Estudios de la Salud, Research and Health Technology Assessment
REVIEW RETURNED	18-Dec-2021
GENERAL COMMENTS	This reviewer considers that the knowledge derived from this study has global implications regarding the size of the mortality associated to COVID19. Since this study is an important observational study, the STROBE checklist is crucial for not

	missing any relevant detail while writing a report. As I mentioned later, it is incomplete as I comment below! Despite using the STROBE checklist, I have some methodological concerns that might hinder the reader to better understand this report. In addition, I strongly recommend the authors, that the manuscript must be proof-read by a native English speaker or a professional proof-reader. 1- The last paragraph of the Introduction section, as required by the STROBE, is the aim of the study. The aim and the methods must be redacted in past tense. 2-There are three observational study designs: cross-sectional, cohort and case-control. Which study design does the best suit the current study? This study must fit one of them! In the previous review, this reviewer requested such information and the authors still do not describe it anywhere in the manuscript. Furthermore, selecting the study design means that some points in the STROBE list require to be checked! 3-The following sentence at the beginning of the methods section "We using weekly death data from the Sistema Informatico Nacional de Defunciones (SINADEF) at the Ministry of Health of Peru" must be rewritten as follows: "We used weekly death data extracted from the National Information Technology System for Death Records (in Spanish 'Sistema Informático Nacional de Defunciones' SINADEF) at the Ministry of Health of Peru" 4-Dates of the month in English language must be written as ordinal numbers: "...from the 1st of January though the 31st of December." In addition, the date of the month must be written before the name of the month in the British English, but this order is reversed in US English -as it is written in the Result section-. The authors must write it using the spelling requested by this journal style. The authors can not use both English spellings in the same manuscript. Furthermore, I strongly encourage the authors to use a professional proofreader or getting advice from native English speaker for improving the writing of this report. 5-Why was the data converted in to ISO weeks? The authors only mention this statement once, later on in the methods section, they refer to the "epidemiological weeks", How different are the "ISO weeks" from the "epidemiological weeks"? What does they mean with such acronym. Acronyms must always be defined before first use and used afterwards. 6-I am a bit confused regarding the data contained in SINADEF: "Open access data includes ... the basic cause of death ... the International Classification of Diseases 10th Revision (ICD-10)". Is the basic cause of death classified according to ICD-10? What other uses does the ICD-10 have besides classifying the basic and additional causes of death when writing a death certificate? 7-It very hard to follow the writing of the "excess mortality analysis" section, because it does not follow a logical order: you start from the description and build up to inferential statistics. First of all, it is customary that such section is called "Statistical Analysis". The use of "P-score" is very confusing for readers, since it looks like a
--	--

	"p value" which is not what Palacio-Mejía et al. meant which is a "proportion of mortality count excess" or something like that. 8-The definition of the number of expected deaths is crucial in this study, because it seems to be inconsistent and even poorly explained in the literature. As far as this reviewer understands, it is "the weekly average number of deaths by sex and age in the three years preceding the pandemics", if this is true, you define it as a variable once and use the term "expected number of deaths" in the manuscript. 9-MR and ISO acrynomns must be defined once and used through the manuscript. 10-I studied medicine, more than 15 years ago. I learned that virus, cancer and intracellular microbes produce intracellular antigens that are presented by non-professional cells to CD8+ cells that become memory cells which is the point of almost all COVID19 vaccines but the Chinese and Cubans ones. Is there information in the literature regarding the role of such cells? 11-Most health systems around the world address the pandemics that is still ongoing. Such efforts for contain the COVID19 virus hinder primary and secondary prevention programs against chronic diseases. Is there information in the literature regarding this issue and is it related to the findings of this study? 12-In the authors mention mortality rates are "asynchronous temporal (which is a redundant term) and geographical distribution" which this reviewer doesn't really know what they mean. As far this reviewer poorly understood, peaks seem to be different among different regions. If it has been the case, why is it not presented in the Results section? 13-Regarding the "Strengths and limitations of this study", please remember that "P-score" is confusing, furthermore this reviewer doesn't understand what do the authors mean when they refer regarding "more sophisticated analyses"? Please consider this before rewriting all this section again and give the reader a clearer message regarding the most important findings of this manuscript.
--	---

VERSION 2 – AUTHOR RESPONSE

Reviewer #1. This reviewer considers that the knowledge derived from this study has global implications regarding the size of the mortality associated to COVID19. Since this study is an important observational study, the STROBE checklist is crucial for not missing any relevant detail while writing a report. As I mentioned later, it is incomplete as I comment below!

Despite using the STROBE checklist, I have some methodological concerns that might hinder the reader to better understand this report. In addition, I strongly recommend the authors, that the manuscript must be proof-read by a native English speaker or a professional proof-reader.

Response: Thank you for your comments. The text has been checked for correct use of grammar and common technical terms, and edited to a level suitable.

We certify that the following article

Excess all-cause deaths stratified by sex and age in Peru: a time series analysis during the COVID-19 pandemic

Max Carlos Ramírez-Soto

has undergone English language editing by MDPI. The text has been checked for correct use of grammar and common technical terms, and edited to a level suitable for reporting research in a scholarly journal.

MDPI uses experienced, native English speaking editors. Full details of the editing service can be found at
► <https://www.mdpi.com/authors/english>.

Comment 1: The last paragraph of the Introduction section, as required by the STROBE, is the aim of the study. The aim and the methods must be redacted in past tense.

Response 1: Thank you for your comments. The sentence was corrected "In this study, we estimated the excess all-cause deaths and the excess death rate during the first year of the COVID-19 pandemic in 25 Peruvian regions, stratified by sex, age, group, and epidemiological week".

Comment 2: There are three observational study designs: cross-sectional, cohort and case-control. Which study design does the best suit the current study? This study must fit one of them! In the previous review, this reviewer requested such information and the authors still do not describe it anywhere in the manuscript. Furthermore, selecting the study design means that some points in the STROBE list require to be checked!

Response 2: Thank you for your comments. We include the study design "We performed a cross-sectional study following the Strengthening the Reporting of Observational Studies in Epidemiology (STROBE) reporting guidelines".

Comment 3: The following sentence at the beginning of the methods section "We using weekly death data from the Sistema Informatico Nacional de Defunciones (SINADEF) at the Ministry of Health of Peru" must be rewritten as follows: "We used weekly death data extracted from the National Information Technology System for Death Records (in Spanish 'Sistema Informático Nacional de Defunciones' SINADEF) at the Ministry of Health of Peru".

Response 3: Thank you for your comments. The sentence was corrected "We used weekly death data extracted from the National Information Technology System for Death Records (in Spanish 'Sistema Informático Nacional de Defunciones' SINADEF) at the Ministry of Health of Peru".

Comment 4: Dates of the month in English language must be written as ordinal numbers: "...from the 1st of January though the 31st of December." In addition, the date of the month must be written before the name of the month in the British English, but this order is reversed in US English -as it is written in the Result section-. The authors must write it using the spelling requested by this journal style. The authors can not use both English spellings in the same manuscript. Furthermore, I strongly

encourage the authors to use a professional proofreader or getting advice from native English speaker for improving the writing of this report.

Response 4: Thank you for your comments. It was corrected (see Methods and Results section).

Comment 5: Why was the data converted in to ISO weeks? The authors only mention this statement once, later on in the methods section, they refer to the "epidemiological weeks", How different are the "ISO weeks" from the "epidemiological weeks"? What does they mean with such acronym. Acronyms must always be defined before first use and used afterwards.

Response 5: Thank you for your comments. This sentence was deleted.

Comment 6: I am a bit confused regarding the data contained in SINADEF: "Open access data includes ... the basic cause of death ... the International Classification of Diseases 10th Revision (ICD-10)". Is the basic cause of death classified according to ICD-10? What other uses does the ICD-10 have besides classifying the basic and additional causes of death when writing a death certificate?

Response 6: Thank you for your comments. It was corrected "Open access data includes information stratified by age/sex, the basic cause of death by the International Classification of Diseases 10th Revision (ICD-10),....".

Comment 7: It very hard to follow the writing of the "excess mortality analysis" section, because it does not follow a logical order: you start from the description and build up to inferential statistics. First of all, it is customary that such section is called "Statistical Analysis". The use of "P-score" is very confusing for readers, since it looks like a "p value" which is not what Palacio-Mejía et al. meant which is a "proportion of mortality count excess" or something like that.

Response 7: Thank you for your comments. This paragraph was corrected "We estimated the excess deaths and the excess death rates (per 100,000 inhabitants) according to sex and age, as well as in each region by sex. Expected deaths in 2020 were obtained from the average number of deaths over the years 2017–2019. The weekly average number of deaths by sex and age in the three years preceding the pandemic was the expected number of deaths. Observed deaths in 2020 were the deaths reported from 1st of January through to the 31st of December. Excess all-cause deaths during the pandemic period were estimated as the difference between observed deaths and expected deaths in 2020. To make comparisons with other countries, we measured excess proportional mortality (proportion of mortality count excess; weekly or monthly deaths in 2020) as the percentage difference between the observed and expected number of deaths ((observed deaths–expected deaths)/expected deaths*100%),⁸ by sex, age, region, and epidemiological week".

Comment 8: The definition of the number of expected deaths is crucial in this study, because it seems to be inconsistent and even poorly explained in the literature. As far as this reviewer understands, it is "the weekly average number of deaths by sex and age in the three years preceding the pandemics", if this is true, you define it as a variable once and use the term "expected number of deaths" in the manuscript.

Response 8: Thank you for your comments. This paragraph was corrected "Expected deaths in 2020 were obtained from the average number of deaths over the years 2017–2019. The weekly average number of deaths by sex and age in the three years preceding the pandemic was the expected number of deaths. Observed deaths in 2020 were the deaths reported from 1st of January through to the 31st of December".

Comment 9: MR and ISO acrynomys must be defined once and used through the manuscript.

Response 9: Thank you for your comments. MR acrynom was defined "Mortality rate".

Comment 10: I studied medicine, more than 15 years ago. I learned that virus, cancer and intracellular microbes produce intracellular antigens that are presented by non-professional cells to CD8+ cells that become memory cells which is the point of almost all COVID19 vaccines but the Chinese and Cubans ones. Is there information in the literature regarding the role of such cells?

Response 10: Thank you for your comments. Information included has been incorporated that mentions the need for further investigation "Because COVID-19 is a new infectious disease, the development and viability of memory T cells in men and women are still unknown, especially in the face of viral mutation. Therefore, further studies could define the sex-differential role of T cells in acute disease. Sex differences in immunopathogenesis could inform mechanisms of COVID-19, and identify points for treatment and increase vaccine efficacy to target parts of the virus that are less likely to mutate, supported by genome analysis.^{26,27}".

Comment 11: Most health systems around the world address the pandemics that is still ongoing. Such efforts for contain the COVID19 virus hinder primary and secondary prevention programs against chronic diseases. Is there information in the literature regarding this issue and is it related to the findings of this study?

Response 11: Thank you for your comments. There is information on perspectives and recommendations to restore primary or secondary prevention programs during the pandemic. The wording includes recommending future studies "Finally, during the country-wide lockdown implemented in Peru, access to medical and surgical care was limited and interrupted across primary and secondary prevention programs. Further studies could evaluate the indirect effects of confinement and excess deaths and mortality by disaggregating data by chronic or acute diseases".

Comment 12: In the authors mention mortality rates are "asynchronous temporal (which is a redundant term) and geographical distribution" which this reviewer doesn't really know what they mean. As far this reviewer poorly understood, peaks seem to be different among different regions. If it has been the case, why is it not presented in the Results section?

Response 12: Thank you for your comments. This has been described in the Results and Discussion sections.

- Results. Figure 6 shows that the rate of excess deaths increased at the beginning of March and reached a peak at the end of the months of May and August. There were different peaks among different regions. Some regions showed staggering increases, with rates of excess deaths reaching 500% in men in Loreto in week 18. The rate of excess deaths decreased by week 52 (Figure 7).
- Discussion. Overall, there were different peaks among different regions of excess mortality.

Comment 13: Regarding the "Strengths and limitations of this study", please remember that "P-score" is confusing, furthermore this reviewer doesn't understand what do the authors mean when they refer regarding "more sophisticated analyses"? Please consider this before rewriting all this section again and give the reader a clearer message regarding the most important findings of this manuscript.

Response 13: Thank you for your comments. The sentence was corrected "Although our analysis was based in proportion of mortality count excess (%) and the excess death rates (per 100,000 inhabitants), our estimates of excess all-cause deaths are similar to estimates of modeled studies".

VERSION 3 – REVIEW

REVIEWER	Quintana, Hedley Instituto Conmemorativo Gorgas de Estudios de la Salud, Research and Health Technology Assessment
REVIEW RETURNED	11-Feb-2022
GENERAL COMMENTS	I have no additional comments